# Reliability and Validity of the Activity Diversity Questionnaire for Older Adults in Japan

**DOI:** 10.3390/ijerph17072384

**Published:** 2020-03-31

**Authors:** Junta Takahashi, Hisashi Kawai, Hiroyuki Suzuki, Yoshinori Fujiwara, Yutaka Watanabe, Hirohiko Hirano, Hunkyung Kim, Kazushige Ihara, Kaori Ishii, Koichiro Oka, Shuichi Obuchi

**Affiliations:** 1Tokyo Metropolitan Institute of Gerontology, Tokyo 173-0015, Japan; junta@tmig.or.jp (J.T.); hkawai@tmig.or.jp (H.K.); suzukihy@tmig.or.jp (H.S.); fujiwayo@tmig.or.jp (Y.F.); h-hiro@gd5.so-net.ne.jp (H.H.); kimhk@tmig.or.jp (H.K.); 2Faculty of Sport Sciences, Waseda University, Saitama 359-1192, Japan; ishiikaori@waseda.jp (K.I.); koka@waseda.jp (K.O.); 3Gerodontology, Department of Oral Health Science, Faculty of Dental Medicine, Hokkaido University, Hokkaido 060-8586, Japan; 4Department of Social Medicine, Hirosaki University Graduate School of Medicine, Aomori 036-8560, Japan; ihara@hirosaki-u.ac.jp

**Keywords:** activity diversity, questionnaire development, community-dwelling, elderly, reliability and validity

## Abstract

Recent research has suggested that the breadth and evenness of activity and activity diversity contribute to health outcomes among older adults. However, few established assessment tools for activity diversity have been developed. This study developed an Activity Diversity Questionnaire (ADQ) for older adults through expert consultation and a preliminary survey among 18 community-dwelling older adults. The diversity score was calculated according to Shannon’s entropy. In study 1, the intraclass correlation coefficients (ICC) of the diversity scores were determined for community-dwelling older adults (*n* = 30). In study 2, concurrent validity was tested with participants receiving comprehensive health checkups at the Itabashi ward in Japan in 2018 (*n* = 766). The correlation coefficients of the diversity scores were then calculated in reference to the Tokyo Metropolitan Institute of Gerontology Index of Competence and Japan Science and Technology Agency Index of Competence. The final version of the ADQ consisted of 20 total items with excellent test-retest reliability (ICC = 0.84) and moderate correlations with both the Tokyo Metropolitan Institute of Gerontology Index of Competence and Japan Science and Technology Agency Index of Competence (r = 0.48 and 0.60, respectively). The ADQ was developed through scientific procedures and revealed sufficient reliability and validity. As such, it is a scientifically validated tool for assessing activity diversity among older adults.

## 1. Introduction

Populational aging is a concern worldwide, especially in well-developed countries, and age-related conditions and disabilities are burdens for the individuals themselves, their families, and public health care systems [1]. Further, increases in the healthy life expectancy rates have been smaller than those in the overall life expectancy rates, and unhealthy life expectancy rate is increasing in most countries [2]. Thus, it is necessary to develop ways to extend the healthy life expectancy rate and to contribute to older adults’ well-being.

Daily activity is widely considered as an important factor for delaying deterioration in functions among older adults. The associations between various types of activities and health among older adults have therefore been intensively studied. Studies have shown that leisure-time, productive, social, and intellectual activities have positive effects on physical, mental, and cognitive health as well as life expectancy [3,4,5,6,7,8,9].

Defined by the total number of activities, activity diversity contributes to well-being and health [10], cognitive function [11], and alleviating depression [12]. These are important factors for health outcomes among older adults. However, previous studies have solely examined the simple sums of these activities. These studies did not consider whether subjects participated in many activities evenly or in limited activities convergently.

Lee et al. defined the breadth (number of activities) and evenness of activities as activity diversity using Shannon’s entropy to examine the relationship between activity diversity and health outcomes among older adults [13,14]. Though limited, Shannon’s entropy is applied in a variety of areas, including stressor diversity [15], emotional diversity [16], and social diversity [17]. Results have shown that individuals with greater activity diversity exhibit better self-perceived and mental health. Further, activity diversity is an important factor for health outcomes among older adults. In this context, it is necessary to define diversity to include not only the breadth of implemented activities, but also the evenness. Additional research is needed to examine the relationship between activity diversity and health outcomes.

However, to our knowledge, there are currently no established methods for assessing activity diversity [13]. Although Lee et al. investigated diversity through seven activities (paid work, engaging with children, doing chores, leisure, physical activities, formal volunteering, giving informal help to people, and intellectual activity) [13], this range may be too small to sufficiently reflect individual differences in activity diversity. In addition, the assessment method for determining activity diversity used in this study was not scientifically validated. It is thus necessary to develop scientifically validated methods for assessing activity diversity based on a sufficient variety of activities that reflect the general daily activities of older adults. The present study, therefore, developed an assessment tool for activity diversity using a sufficient variety of activities. The tool was then scientifically verified for validity and reliability.

## 2. Materials and Methods

### 2.1. Development of the Activity Diversity Questionnaire

The pilot version of the Activity Diversity Questionnaire (ADQ) was prepared in consultation with expert researchers (i.e., two in gerontology, one in health and behavioral sciences, and one in sociology) who were familiar with issues related to older adults. The activities used in the pilot version of the ADQ were selected by referring to previous studies [3,18,19] that investigated general activities among older adults. Next, a preliminary survey to investigate the adequacy of the pilot version was conducted among community-dwelling older adults aged 65 years and above who were participants of exercise class at our institute. These classes were conducted twice a week to promote participants’ health, and approximately 30 older residents residing near the institute attended the classes. We explained the study details to the potential participants during a class and requested their cooperation. A total of 18 participants were recruited (four men and 14 women). The participants were assessed using the pilot version of the ADQ and asked whether they performed any daily activities other than the items in the pilot version. The questionnaire was then revised by modifying or adding items based on these results. 

### 2.2. Calculating the Diversity Score

The ADQ asked participants the following question: “How many times did you perform the following activities during the past week?” Each listed item was answered using the following scale: “0: almost never,” “1: 1–2 days per week,” “2: once every 2 days,” and “3: almost every day.” Frequency scores of 0 to 3 were thus allocated to each item. These frequency scores were then used to calculate a diversity score according to the following formula [14].
$$Diversity\;score=-\left(\frac{1}{logm}\right)\overset{m}{\underset{i=1}{\sum }}P_{i}\left(logP_{i}\right)$$
where *m* = 20 is the number of activity types, *i* is each activity (*i* = 1 to *m*), and *P_i_* is *i*’s frequency proportion of total activities (= *i*’s frequency score/total frequency score from 0 to m). A diversity score ranges from 0 to 1, with higher scores indicating more numbers of activity types and spreading evenly across all 20 categories.

### 2.3. Study 1—Test-Retest Reliability Verification

#### 2.3.1. Participants

Participants were 30 community-dwelling older adults who belonged to local self-exercise groups at the Itabashi wards and Shinjuku wards, Tokyo, Japan. The inclusion criteria were community-dwelling individuals aged 65 years or older who were without dementia. The participants were asked, “Have you had any changes (e.g., moving, feeling sick, traveling) in your life in the last week?” and were required to answer “no change,” “slight change,” or “big change”. Those who answered “big change” were excluded. All participants were provided the study details and the advantages and disadvantages of participating. All participants provided informed consent before participating.

#### 2.3.2. Protocol

The participants were required to answer the ADQ by themselves on two separate occasions, each separated by one week. Demographic information was taken during the first instance, including age, sex, and education history. The study protocol was approved by the ethics committee at the Waseda University (approval number: 2019-095).

#### 2.3.3. Statistical Analysis

The intraclass correlation coefficients (ICC) of the diversity score at the first and second administrations were determined. The ICC were calculated using a two-way mixed effects model. The ICC was interpreted as indicating poor reliability (<0.4), fair to good reliability (≥0.4 to <0.75), and excellent reliability (≥0.75) [20]. All analyses were conducted using IBM SPSS version 23.0 (IBM Japan, Tokyo, Japan). Calculated p-values < 0.05 were statistically significant.

### 2.4. Study 2—Concurrent Validity Verification

#### 2.4.1. Participants

The participants were individuals who had undergone comprehensive health checkups in 2018 at the Tokyo Metropolitan Geriatric Hospital and Institute of Gerontology (Tokyo, Japan) as part of the Otassha Kenshin study, which was initiated in 2011 and continued for one decade; the details of the study have been described in a previous article [21,22]. The inclusion criteria were community-dwelling older individuals aged 65 years or older and who were without dementia. All participants were given sufficient information about the purpose of the research and before participating.

#### 2.4.2. Measurements

We used the Tokyo Metropolitan Institute of Gerontology Index of Competence (TMIG-IC) and Japan Science and Technology Agency Index of Competence (JST-IC) as external standards [23,24]. Although both indexes measure higher-level competence in Lawton’s seven-stage model of human behavior [25], the JST-IC assesses higher competences than the TMIG-IC [23]. Other information was also taken during the survey, including age, sex, education, medical history (cerebral apoplexy, heart disease, diabetic and depression), subjective economic status, household composition, and the ADQ.

#### 2.4.3. Statistical Analysis

The correlations between diversity scores derived using the ADQ and those from the TMIG-IC and JST-IC were determined using Pearson’s correlation coefficient. Pearson’s correlation was interpreted as insubstantial (<0.1), small (≥0.1 to <0.3), moderate (≥0.3 to <0.50), and large (≥0.50 [26]. All analyses were conducted using IBM SPSS version 23.0. Calculated p-values < 0.05 were statistically significant. This study was approved by the ethics committee at the Tokyo Metropolitan Institute of Gerontology (approval number: 28-3549).

## 3. Results

### 3.1. Developing the Activity Diversity Questionnaire

The initial pilot version of the ADQ contained a total of 18 items. The two items of “Shopping” and “Listening to the radio” were added after conducting a preliminary survey among community-dwelling older adults. An existing item was also modified; because some participants drove motorcycles, the item about cars was amended to “Driving cars and motorcycles.” Table A1 shows the final version of the ADQ consisted of 20 total items (Inside chores, Outside chores, Grooming, Screen time, Listening to the radio, Playing games, Gambling, Shopping, Direct contact with friends/relatives, Indirect contact with friends/relatives, Leisure activities with physically demanding components, Leisure activities without physically demanding components, Regional activities/volunteering, Working, Childcare, Older care, Pet care, Hospital attendance, Driving cars/motorcycles, and Public transportation usage). All participants were sufficiently explained about the ADQ at the beginning. None of the participants reported any difficulty responding to the ADQ during the preliminary survey.

### 3.2. Study 1—Test-Retest Reliability Verification

In total, there were 30 participants (all female), median age 77 (range 69–92) years, mean years of education (standard deviation; SD) was 11.7 (1.9) years; none had undergone any major changes in daily life during the examination period. Table 1 shows the diversity scores and ICCs for the first and second administrations of the ADQ. The diversity scores of the first and second administration were 0.73 (range 0.23–0.86) and 0.76 (range 0.43–0.90), the ICC of diversity scores was excellent (0.84, *p* < 0.01).

### 3.3. Study 2—Concurrent Validity Verification

Participant characteristics are shown in Table 2. As seen, there were 766 total participants (female 60.4%), median age was 73.5 (range 65–91) years. The ADQ indicated that the total number of activities performed over the past week was 10.5 (2.5; range 3–18); the total frequency score was 22.9 (5.5; range 5–40); the diversity score was 0.74 (0.09; range 0.32–0.93). Table 3 shows the correlations between the diversity scores derived from the ADQ and those taken from the TMIG-IC and JST-IC (correlation coefficients were 0.48 (*p* < 0.01) and 0.60 (*p* < 0.01), respectively).

## 4. Discussion

### 4.1. Study Significance

This study developed the ADQ and verified its test-retest reliability and concurrent validity. The ADQ was constructed through two procedures, including consultations with gerontology experts and a preliminary survey among community-dwelling older adults. This questionnaire defined activity diversity using Shannon’s entropy, and was the first assessment tool of activity diversity that had been verified for reliability and validity. Activity diversity including not only breadth, but also evenness, which may be more significantly associated with the health outcomes of older adults than either the simple sum of activities in which individuals engage or physical activity measurements. This study is fundamental to the future investigation of the association between activity diversity and health outcomes among older adults.

### 4.2. Content Validity

The activities in the ADQ were selected through expert consultation and preliminary surveys; nearly all activity domains were included (personal leisure, civic/religious activity, physical exercise, interior household chores, exterior household chores, managing medical conditions, employment/computer use, interpersonal exchange/helping others, community leisure) based on previous research [19]. The ADQ also uniquely included the element of gambling. This is because a previous study indicated that approximately 20% of all Japanese adults engaged in gambling activities [27], which are very common throughout Japan [28]. The ADQ was designed to include this Japanese cultural characteristic in order to more accurately consider general daily activity among community-dwelling older adults in that context. From the above, the ADQ can be considered an index that sufficiently reflects general and wider daily activities among Japanese community-dwelling older adults. These processes also yielded content validity.

### 4.3. Test-Retest Reliability

The ICC between the two administrations was 0.84, indicating excellent reliability [29]. The findings suggest that the ADQ had acceptable test-retest reliability for use among community-dwelling older adults. Although a previous study recommended 12.9 days as the time interval for investigating test-retest reliability in patient-reported outcome measures [30], the time interval in this study was maintained at 7 days because daily activity appeared to be sensitive to changes in individual physical or mental condition, families, friends, and the season. In fact, no participants underwent major life changes during the examination period. As such, the time interval seemed reasonable.

### 4.4. Concurrent Validity

The correlation coefficients of the diversity scores between the ADQ and the TMIG-IC and JST-IC scores were 0.48 and 0.60, respectively, thus showing moderate correlations. A previous study reported that the correlation coefficient between the Motor Fitness Scale (an index of physical fitness) [31] and the JST-IC was 0.64; the ADQ showed similar results. As the TMIG-IC and JST-IC are designed to assess higher-level competence in regard to human behavior among older adults [24,32], we believe that their assessment concepts were highly similar to that of the ADQ. Therefore, these indexes seemed reasonable as external standards. The TMIG-IC is related to certain health outcomes among older adults, including care needs, long-term care costs, and mortality [23,33,34]. Activity diversity may be associated with these health outcomes.

### 4.5. Study Limitations and Perspectives

This study also has some limitations. First, there may have been a recall bias because the ADQ was in a questionnaire form. The second limitation is the possible selection bias in recruiting participants. The participants in Study 1 may not represent the community-dwelling older adults, because the participants were female and member of a self-exercise group. Moreover, since Study 2 was conducted as a venue survey, only those who were able to arrive at the predetermined location by themselves were able to participate. Finally, the questionnaire was only validated within a timeframe of seven days. Considering that this could result in false assumptions if the questionnaire is administered for longer survey periods, a longer-term survey is needed.

Future studies should examine the differences in characteristics between individuals with high activity diversity and individuals with low activity diversity, the relationship between activity diversity and health outcomes such as mortality, non-communicable disease, and frailty. Additionally, the cross-cultural validity of the ADQ needs to be examined in other countries.

## 5. Conclusions

This study defined the breadth and evenness of daily activities among older adults to determine activity diversity using Shannon’s entropy. Activity diversity is a novel index of health behavior that differs from other indices of health behavior, such as type and frequency of daily activities. The ADQ was developed through scientific procedures (e.g., expert consultation and a preliminary survey) and examined for both test-retest reliability and concurrent validity. The results showed sufficiently reliability and validity, indicating that the ADQ is a useful tool to assess activity diversity. ADQ may be a better measure of daily activity in older adults without being affected by specific lifestyle and physical activity. We believe that activity diversity is an important factor for health outcomes among older adults and requires continued scholarly investigation.

## Figures and Tables

**Table 1 ijerph-17-02384-t001:** Intraclass correlation coefficients (ICC) of the Activity Diversity Questionnaire.

	First Administration		Second Administration		Test-Retest Reliability				*p* Value
	Mean	Standard Deviation	Mean	Standard Deviation	ICC	95% Confidence Interval			
**Diversity score**	0.73	(0.14)	0.76	(0.11)	0.84	0.63	-	0.93	<0.01

**Table 2 ijerph-17-02384-t002:** Participant characteristics.

	Mean (Standard Deviation)/N (%)	
Median age in years (range)	73.5	(65–91)
Sex (female)	463	(60.4)
Education in years	13.4	(2.8)
Medical history		
Cerebral apoplexy	51	(6.7)
Heart disease	129	(16.8)
Diabetic	90	(11.7)
Depression	32	(4.2)
Subjective economic status		
Sufficient	653	(85.2)
Insufficient	113	(14.8)
Household composition		
With housemate(s)	562	(73.4)
Living Alone	204	(26.6)
TMIG-IC ^a^	11.9	(1.7)
JST-IC ^b^	12.0	(2.9)
Diversity score	0.74	(0.09)

*Note*. ^a^ TMIG-IC: Tokyo Metropolitan Institute of Gerontology Index of Competence, ^b^ JST-IC: Japan Science and Technology Agency Index of Competence.

**Table 3 ijerph-17-02384-t003:** Correlation coefficient between external standards and the diversity score.

	TMIG-IC ^a^		JST-IC ^b^	
	*r*	*P*	*r*	*P*
Diversity score	0.48	<0.01	0.60	<0.01

*Notes*. ^a^ TMIG-IC: Tokyo Metropolitan Institute of Gerontology Index of Competence, ^b^ JST-IC: Japan Science and Technology Agency Index of Competence.

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
