# Peer review of "Reliability and Validity of the Activity Diversity Questionnaire for Older Adults in Japan"

_ijerph, 2020, doi:10.3390/ijerph17072384_

Round 1

Reviewer 1 Report

Introduction

In my view the introduction is well written, but here and there a bit too wordy, e.g. "huge" concern; "Indeed"; Daily "life" activity...

Line 37-38: To my knowledge, Japan has one of the highest life expectancy rates world wide. Not surprisingly the increase in this rate is slower in comparison to other countries. I do not see the urgency to increase the lifetime of people, but to improve health and well-being in all stages of life. Authors should legimitate precisely their motivation for this work.

Minor grammar mistakes. Consider further proof-reading. e.g. line 39: "widely considered as important factor"

Line 58: This statement needs a proof.

Line 63: So you are saying that there are already some measures existing (in contradiction to line 58), but they are not validated. One or two examples of those studies would illustrate the urgency to develop such a measure.

Line 69: No abbreviation in the header.

Materials and Methods

Line 75: How was the selection process of the sample?

Line 76: draft might not be the correct wording. "first version" How many items were inside the first version?

Line 89: Use math formula for P_i.

Line 90 - 92: Please be more concise and explain the calculation without using brackets.

Line 97: What is defined as major change? An example would be nice.

Line 98: What do you mean by "sufficient" information?

Line 104: Was the study registered? Please add the registration number.

Line 106: I think there are wrong citations. What does (1, 2) mean? Could you please describe how the ICC was calculated? Which formula did you use?

Line 108: Stick to the SAMPL guidelines (https://www.equator-network.org/2013/02/11/sampl-guidelines-for-statistical-reporting/) when reporting p-values, numbers etc.

Considering that two studies are reported in one article I would suggest to rearrange the methods section, so that you do not have to write everything twice.

Results

Line 138: Do you account for activities which can be conducted at the same time? Radio listening and doing chores?

Line 142: Erase (Table S1) at the end of the sentence. Prefer: Table S1 shows the final version of...

Line 142: How did you assess that the participants had no difficulties?

Line 145: Why only women? Please describe the recruitment procedure and maybe discuss a selection bias. Please consider SAMPL guidelines when reporting numbers.

Table 1 and in all following tables. Format brackets of standard deviation values.

Line 154: Consider reporting the proportion of female instead of the number.

Line 154 - 160: Some spaces before numbers are missing. Consider further proof reading regarding this point in the whole manuscript. Especially reporting mean (SD) or mean±SD. See SAMPL guidelines and be consistent.

Table 2: It would be interesting to analyse the effects of the participants characteristcs on their activity diversity. Did you consider an analysis? If not, state that this will be incorporated in future studies.

Line 200: If you consider the questionnaire in future studies you actually need to assess all other influences on the activity behaviour as well. Since the questionnaire is only validated within a timeframe of seven days. This could lead to false assumptions if the questionnaire is administered for longer survey periods.

Conclusion

Line 224: I agree that the questionnaire needs further investigations. Consider a validation with device-based activity measures and in other cultural backgrounds in future studies.

Line 233: "supervised" instead of "oversaw"

Table S1: Answers are not in the same column as reporting numbers. Please format.

References

Consider further proof reading of citation style!

Reference 10: Error "S74-S82"

Author Response

Introduction

In my view the introduction is well written, but here and there a bit too wordy, e.g. "huge" concern; "Indeed"; Daily "life" activity...

⇒Thank you for your suggestion. The following words were deleted for conciseness:

“huge” (line 34), “Indeed” (line 34), Daily “life” activity (line 40).

Line 37-38: To my knowledge, Japan has one of the highest life expectancy rates worldwide. Not surprisingly the increase in this rate is slower in comparison to other countries. I do not see the urgency to increase the lifetime of people, but to improve health and well-being in all stages of life. Authors should legitimate precisely their motivation for this work.

⇒Thank you for your comment. Our study is not aimed at extending life expectancy, but at increasing healthy life expectancy and contributing to people's well-being. I added some details to clarify this point (Lines 37-38).

Minor grammar mistakes. Consider further proof-reading. e.g. line 40: "widely considered as important factor"

⇒Thank you for pointing it out. This has been revised accordingly (Line 40).

Line 59: This statement needs a proof.

⇒To the best of our knowledge, there is no established method to assess activity diversity. I have added a citation (Lines 59-60).

Line 64: So you are saying that there are already some measures existing (in contradiction to line 59), but they are not validated. One or two examples of those studies would illustrate the urgency to develop such a measure.

⇒Since only Lee et al.’s study evaluated activity diversity based on the same definition used in this study, we revised this to its singular form (Line 64).

Materials and Methods

Line 70: No abbreviation in the header.

⇒This has been revised as pointed out (line 70)

Line 75: How was the selection process of the sample?

⇒I added the details of the selection process (Lines 76-80)

Line 75: "first version" How many items were inside the first version?

⇒We modified “draft” to “pilot version” (Line 75). The pilot version of the ADQ consisted of 18 items. This is stated in the results (Line 143)

Line 92: Use math formula for Pi

⇒ We have used the mathematical formula for Pi (Line 93).

Line 93 - 94: Please be more concise and explain the calculation without using brackets.

⇒This has been revised as instructed (Lines 94-95)

Line 100: What is defined as major change? An example would be nice.

⇒I added the following explanation (Lines 100-103)

“The participants were asked, "Have you had any changes (e.g. moving, feeling sick, traveling) in your life in the last week?" and were required to answer "no change," "slight change," or "big change". Those who answered "big change" were excluded.”

Line 103: What do you mean by "sufficient" information?

⇒"Sufficient information" refers to information about the details of this study and the advantages and disadvantages of participating. I added this point. (Lines 103-104)

Line 110: Was the study registered? Please add the registration number.

⇒I added the registration numbers (Lines 110 and 139)

Line 113: I think there are wrong citations. What does (1, 2) mean? Could you please describe how the ICC was calculated? Which formula did you use?

⇒I calculated the ICC using a two-way mixed effects model. I have added this information. (Line 113)

Line 118: Stick to the SAMPL guidelines (https://www.equator-network.org/2013/02/11/sampl-guidelines-for-statistical-reporting/) when reporting p-values, numbers etc.

⇒I have revised these according to the guidelines.

Considering that two studies are reported in one article I would suggest to rearrange the methods section, so that you do not have to write everything twice.

⇒To avoid any confusion between the methods of validity and reliability, I described them separately.

Results

Line 142: Do you account for activities which can be conducted at the same time? Radio listening and doing chores?

⇒No explanation was provided to the participants about the activities that could be conducted at the same time. There were also no questions from the participants regarding this point.

Line 146: Erase (Table S1) at the end of the sentence. Prefer: Table S1 shows the final version of...

⇒I revised as suggested (Lines 146-147).

Line 152: How did you assess that the participants had no difficulties?

⇒We did not objectively assess the difficulty of answering the ADQ, but all participants were able to answer it by themselves, and none complained of the difficulty of answering. I added that (Lines 152-154)

Line 156: Why only women? Please describe the recruitment procedure and maybe discuss a selection bias. Please consider SAMPL guidelines when reporting numbers.

⇒I added a sentence in the limitation that the study included only female participants (Lines 227-229)

Table 1 and in all following tables. Format brackets of standard deviation values.

⇒These have been revised as instructed.

Line 165: Consider reporting the proportion of female instead of the number.

⇒I revised as suggested (Line 164)

Line 164 - 167: Some spaces before numbers are missing. Consider further proof reading regarding this point in the whole manuscript. Especially reporting mean (SD) or mean±SD. See SAMPL guidelines and be consistent.

⇒I revised as suggested.

Table 2: It would be interesting to analyses the effects of the participants characteristics on their activity diversity. Did you consider an analysis? If not, state that this will be incorporated in future studies.

⇒As this study aimed to examine the accuracy of this questionnaire itself, the difference in characteristics due to activity diversity is the next issues.

This is stated in 4.5 Study Limitations and Perspectives. I added that. (line 234-235)

Line 162: If you consider the questionnaire in future studies you actually need to assess all other influences on the activity behaviour as well. Since the questionnaire is only validated within a timeframe of seven days. This could lead to false assumptions if the questionnaire is administered for longer survey periods.

⇒I added this point to limitation (Lines 231-233)

Conclusion

Line 239: I agree that the questionnaire needs further investigations. Consider a validation with device-based activity measures and in other cultural backgrounds in future studies.

⇒Thank you for suggestion. This is an issue to be considered in a future study.

Line 256: "supervised" instead of "oversaw"

⇒I revised as suggested (Line 255)

Table S1: Answers are not in the same column as reporting numbers. Please format.

⇒I revised as suggested.

References

Consider further proof reading of citation style!

⇒Citations in text have been checked as per the journal guidelines.

Reference 10: Error "S74-S82"

⇒I have revised this as suggested. (line 289)

Reviewer 2 Report

This paper offers a unique purpose and has a huge potential. In my opinion the value of this article is its methods.  I hope that my suggestions will help improve a contents and manuscript could have great impact to the readers.

My comments relate primarily to the Material and Methods paragraph.

2.3.1 Participants - what should be described:

  • inclusion critria as well as exclusion criteria (more precisely), 
  • health status of the participants (older adults have usually few health problems)
  • social-economical status and social structure of the examined group (sex, age: mean, sd, level of education etc.)

2.3.2 Statistical analysis

  • why did you chose ICC2,1 model (it should be explained),
  • what were the ranges of values (indicate quotation)
  • line 108: I suggest to use α=0.05 level of significance or write sencence: "calculated p-values <0.05 were statistically significant" p: lower case

2.4 Study 2 - this paragraph has 2.4 number, you have three sub paragraphs in it; why they are again renumbered 2.3.1, 2.3.2, 2.3.3. I suppose it should be 2.4.1, 2.4.2, 2.4.3 respectively.

3. Results

line 150: ICC results should be interpreted according to criteria

Author Response

2.3.1 Participants - what should be described:

inclusion criteria as well as exclusion criteria (more precisely),

health status of the participants (older adults have usually few health problems)

social-economical status and social structure of the examined group (sex, age: mean, sd, level of education etc.)

⇒Exclusion criteria are described in more detail. (line 100-103)

The characteristics of the participants are described in detail in the results (Lines 156-158)

2.3.2 Statistical analysis

why did you chose ICC2,1 model (it should be explained),

⇒I made a mistake in the description. I calculated the ICC using a two-way mixed effects model. I added that. (Line 113)

what were the ranges of values (indicate quotation)

⇒The correlations and ICC parameters were determined according to the criteria of previous studies (Lines 114-115 and Lines 135-136).

line 116: I suggest to use α=0.05 level of significance or write sencence: "calculated p-values <0.05 were statistically significant" p: lower case

⇒I revised as suggested (Lines 116 and 137).

2.4 Study 2 - this paragraph has 2.4 number, you have three sub paragraphs in it; why they are again renumbered 2.3.1, 2.3.2, 2.3.3. I suppose it should be 2.4.1, 2.4.2, 2.4.3 respectively.

⇒I revised as suggested.

  1. Results

line 160: ICC results should be interpreted according to criteria

⇒I revised as suggested (Lines 160).

Reviewer 3 Report

This manuscript presents results of the development and studies of psychometric properties for an instrument that measures activity diversity among older adults in Japan. Each step in the process is provided, from content input from multiple content experts regarding the activities included in the questionnaire, to a preliminary study of older adults to further inform the items included, and studies of test-retest reliability and concurrent validity. The results support the instrument as a valid measure to assess activity diversity among older adults.

The strengths of the manuscript include the use of established methods for the determination of the important psychometric properties of reliability and validity as well as the first development of an instrument to measure an important dimension in the lives of older adults and the quality and potential problems with which it is associated. The large sample size for the concurrent validity test is another strength.

There are some issues that might be addressed.

  • Participants: In the Test-Retest study, the 30 participants “belonged to local self-exercise groups” (line 95). This does not sound like a necessarily representative group of older adults, but rather a convenience sample. Would the authors concede this as a possible issue that might suggest a more general sample of participants in future research? That is, are these older adults possibly healthier, more active, etc. than general population older adults in Japan?
  • Limitation - Culture: The development of the instrument, the selection of concurrent validity measures, etc. were all conducted within Japanese culture and experience. The need for reliability and validity of the measure in other cultures might also be an issue for future research and possible use of the instrument. An example is given by the authors for instance with regard to the addition of gambling to the activities included following the initial development study with the 18 older adults. Would this item be appropriate in other cultures in which the measure might be utilized? Would the addition of other activites be observed in older adults of other cultures?
  • Implications: In the Conclusions (line 219), there is an argument made that activity diversity deserves more “scholarly attention.” A perhaps missing addition to the conclusion would be some suggestion(s) regarding implications, uses, or applications of the concept and thus specific utilization of the ADQ.

Author Response

Participants: In the Test-Retest study, the 30 participants “belonged to local self-exercise groups” (line 98). This does not sound like a necessarily representative group of older adults, but rather a convenience sample. Would the authors concede this as a possible issue that might suggest a more general sample of participants in future research? That is, are these older adults possibly healthier, more active, etc. than general population older adults in Japan?

⇒As stated in your comment, there is a possibility that the participants had better physical and cognitive functions than the general community-dwelling older adults, but it is unclear because no detailed evaluation of physical and cognitive functions has been conducted. The limitation was added (Lines 228-230)

Limitation - Culture: The development of the instrument, the selection of concurrent validity measures, etc. were all conducted within Japanese culture and experience. The need for reliability and validity of the measure in other cultures might also be an issue for future research and possible use of the instrument. An example is given by the authors for instance with regard to the addition of gambling to the activities included following the initial development study with the 18 older adults. Would this item be appropriate in other cultures in which the measure might be utilized? Would the addition of other activities be observed in older adults of other cultures?

⇒Activities included in the ADQ do not include activities conducted only in Japan. Therefore, the ADQ may be used in other countries. This was added to the recommendation that cross-cultural validity be examined in the future. (Line 238)

Implications: In the Conclusions (line 239), there is an argument made that activity diversity deserves more “scholarly attention.” A perhaps missing addition to the conclusion would be some suggestion(s) regarding implications, uses, or applications of the concept and thus specific utilization of the ADQ.

⇒I revised conclusions (Lines 241-242, 245-247)